# Highly efficient CRISPR-Cas9-mediated gene knockout in primary human B cells for functional genetic studies of Epstein-Barr virus infection

Ezgi Akidil [1,2‡], Manuel Albanese [1,2,3‡], Alexander Buschle [1,2], Adrian Ruhle [2,3], Dagmar Pich [1,2], Oliver T. Keppler [2,3], Wolfgang Hammerschmidt [1,2]*

1 Research Unit Gene Vectors, Helmholtz Zentrum München, German Research Center for Environmental Health, Munich, Germany, 2 German Center for Infection Research (DZIF), Partner site Munich, Munich, Germany, 3 Max von Pettenkofer Institute and Gene Center, Virology, National Reference Center for Retroviruses, Faculty of Medicine, LMU München, Munich, Germany

‡ These authors share first authorship on this work.
* hammerschmidt@helmholtz-muenchen.de

**Data Availability Statement:** RNA-seq NGS files are deposited on ArrayExpress using the web-based submission tool Annotare 2.0 (https://www.

## Abstract

Gene editing is now routine in all prokaryotic and metazoan cells but has not received much attention in immune cells when the CRISPR-Cas9 technology was introduced in the field of mammalian cell biology less than ten years ago. This versatile technology has been successfully adapted for gene modifications in human myeloid cells and T cells, among others, but applications to human primary B cells have been scarce and limited to activated B cells. This limitation has precluded conclusive studies into cell activation, differentiation or cell cycle control in this cell type. We report on highly efficient, simple and rapid genome engineering in primary resting human B cells using nucleofection of Cas9 ribonucleoprotein complexes, followed by EBV infection or culture on CD40 ligand feeder cells to drive *in vitro* B cell survival. We provide proof-of-principle of gene editing in quiescent human B cells using two model genes: *CD46* and *CDKN2A*. The latter encodes the cell cycle regulator p16$^{INK4a}$ which is an important target of Epstein-Barr virus (EBV). Infection of B cells carrying a knockout of *CDKN2A* with wildtype and EBNA3 oncoprotein mutant strains of EBV allowed us to conclude that EBNA3C controls *CDKN2A*, the only barrier to B cell proliferation in EBV infected cells. Together, this approach enables efficient targeting of specific gene loci in quiescent human B cells supporting basic research as well as immunotherapeutic strategies.

## Author summary

Human hematopoietic stem cells and their derivatives of the myeloid and lymphoid lineages are important targets for gene correction or modifications using the CRISPR-Cas9 technology. Among others, this approach can support site-specific insertion of chimeric antigen receptors (CARs) or T cell receptors (TCRs) into primary T cells. Their subsequent adoptive transfer to patient donors is a promising immunotherapeutic concept that

ebi.ac.uk/fg/annotare/login/). The data files of interest can be downloaded via https://www.ebi.ac.uk/arrayexpress/ using the following accession numbers: RNA-seq: https://www.ebi.ac.uk/arrayexpress/experiments/E-MTAB-9531

**Funding:** This work was financially supported by grants from the Deutsche Krebshilfe (grant number 70112875), German Center for Infection Research (Deutsches Zentrum für Infektionsforschung, DZIF, grant number 07.814) and Deutsche Forschungsgemeinschaft (grant number SFB1064/TP A13) to W.H. The funders had no role in study design, data collection and analysis, decision to publish, or preparation of the manuscript.

**Competing interests:** The authors have declared that no competing interests exist.

may control chronic infection or certain types of cancer. Human B cells have a similar potential but, in contrast to T cells, they are very sensitive, difficult to handle, and short-lived *ex vivo* precluding their genetic modification. Here, we provide efficient means to manipulate primary human B cells genetically using *in vitro* assembled Cas9 ribonucleo-protein complexes. Subsequently, we used Epstein-Barr virus (EBV) infection to ensure the cells' *in vitro* survival for long-term investigations. Our study demonstrates near-to-complete loss of a model target gene and provides examples to evaluate a cellular gene with a critical role during viral infection.

## Introduction

The CRISPR-Cas9 technology has developed into an essential tool for gene editing and genome manipulation in many eukaryotic cell types [1]. This versatile system has been successfully applied to various mammalian cells and cell lines, but its application in certain primary human cells is far from routine. The technology has its limits when it comes to studying the biology of cells that cannot be easily maintained in culture *ex vivo* such as lymphocytes and other immune cells.

Early studies that applied CRISPR-Cas9 mediated gene editing in primary human T cells resulted in low targeting efficiencies and cellular toxicity [2,3] probably due to the delivery of the Cas9 encoding plasmid DNA together with the two indispensable RNAs to assemble a functional CRISPR-Cas9 complex in the cells. *In vitro* assembly of purified bacterial Cas9 nuclease with synthetic CRISPR RNA (crRNA) and trans-activating CRISPR RNA (tracrRNA) followed by nucleofection of the ribonucleoprotein (RNP) complex directly into cells was a major breakthrough in the T cell field [4,5]. The highly efficient knockout of a target gene in primary human T cells proved that the RNP-based CRISPR-Cas9 technology can be applied to develop T cell based immunotherapies and suggested that it might be adaptable to other primary cell types [6].

In contrast to T cells, primary B cells from humans thus far have received little attention in the field of efficient gene manipulation even though they are involved in numerous biological processes, autoimmune and infectious diseases. Until now, several studies using the CRISPR-Cas9 technology have focused on mouse B cells or immortalized human B cell lines latently infected with Epstein-Barr virus (EBV) [7]. Although high-efficiency gene knockouts in B cells have been achieved in these established models, there clearly remains the need to edit the genome of resting primary human B cells. Along these lines, three independent groups introduced RNP complexes into human B cells that had been pre-activated with a cocktail of cytokines [8–10]. Results from these studies suggested that the efficiency of gene editing correlated with the level of B cell stimulation prior to nucleofection of the RNP complexes.

It is widely accepted that resting primary human B cells are refractory to genetic manipulation. Transfection of DNA is extremely inefficient [11] and gene vectors based on retroviruses, lentiviruses or adenoviruses reach transduction rates of <1% in these cells even when we follow published protocols [12,13]. This refractory state is probably due to the cells' very reduced transcriptional activity, unknown restrictions that prevent efficient viral transduction, and the cells' rapid apoptosis within a couple of hours upon cultivation *in vitro* [14]. In addition, guide RNA (gRNA) directed Cas9 endonuclease induces double stranded breaks (DSBs) at the target sites that may cause a potentially fatal DNA damage response (DDR) in B cells. DSBs might be yet another problem, because it is uncertain whether endogenous DNA repair mechanisms are

active in resting cells to mend DSBs adequately. To our knowledge, cellular repair functions of damaged DNA have not been studied in primary human B cells.

In contrast to other viruses, Epstein-Barr virus and its vector derivatives are very proficient reaching high infection and transduction rates in resting primary B cells [15]. The lack of efficient gene editing has been a major roadblock for studies into human B cell biology and in particular for research on EBV. While large panels of EBV mutant viruses are readily available, studies into genetic variants of EBV's genuine human target cells, mature B cells, have not been possible before. Here, we report on highly efficient genome engineering in primary human B cells using the CRISPR-Cas9 technology. *In vitro* assembled RNPs can be delivered to primary resting human B cells by nucleofection followed by EBV infection to study virus biology and maintain the cells long-term.

Our manuscript demonstrates gene editing of the *CD46* locus reaching efficiencies of 85% and beyond even when the primary human B cells were left uninfected. Studies into the kinetics of DSBs in these cells revealed that Cas9 introduced DSBs very rapidly at the *CD46* locus in the majority of cells within hours after nucleofection. Next generation sequencing of CD46 mRNAs metabolically labeled with 4-thiouridine (4sU) documented gene editing and locus repair within 24h after nucleofection indicating the rapid clearance and active transcription of the edited gene locus.

For a functional proof-of-principle, we targeted an EBV relevant cellular gene, *CDKN2A*, which encodes p16$^{INK4a}$, the CDK4 kinase and cell cycle inhibitor. Our findings confirmed a previous report [16] documenting that p16$^{INK4a}$ is an important target of the viral EBNA3C repressor protein and the only barrier to EBV-induced B cell proliferation. In a counterexample, our experiments did not recapitulate the function of two viral transcriptional repressors, EBNA3A and EBNA3C, which were reported to block terminal differentiation of mature B cells into plasmablasts or plasma cells [17].

## Results

### CRISPR-Cas9 efficiently targets the locus of the cell surface protein CD46 in primary human B cells

We aimed at testing the functional knockout of genes in primary human B cells with the CRISPR-Cas9 technology. Towards this end, we designed Cas9-gRNA ribonucleoprotein complexes to target exon 2 of the *CD46* gene. This gene encodes a cell surface protein present on almost all cells, which is also robustly expressed in mature human B cells [18](http://ebv-b.helmholtz-muenchen.de/). This cell surface protein can protect cells from complement attack, it modulates adaptive immune responses in T cells and is a receptor exploited by certain bacteria and viruses. Despite these diverse functions, the survival of human B cells does not depend on its expression even *in vivo* [19].

The recombinant *Streptococcus pyogenes* Cas9 nuclease was assembled *in vitro* using a two-component gRNA protocol with tracrRNA and crRNA. RNP complexes with two different gRNAs (gRNA1, gRNA2) were used individually or in combination. These gRNAs target two sites in the second exon of the gene locus that are 98 bp apart. As shown in **Fig 1A**, the two CD46-Cas9 RNP complexes were nucleofected into primary human B cells purified from human adenoid biopsies as described in Materials and Methods. Since primary human B cells rapidly undergo apoptosis when cultured *ex vivo* in the absence of stimulatory factors, we infected the cells with wild-type (WT) EBV at an optimal multiplicity of infection [14] one hour after nucleofection. EBV infection rescues the *ex vivo* cultivated B cells from cell death and reprograms them to emerge as proliferating lymphoblastoid cell lines [18]. The efficiency of gene editing was investigated both at the level of nucleotide sequence and surface protein

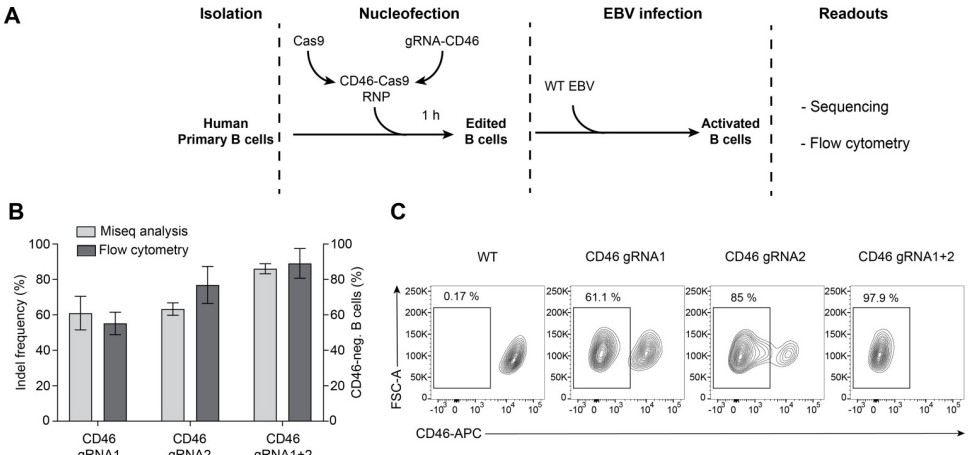

**Fig 1. Targeting the locus of the cell surface protein CD46 in primary human B cells. (A)** Schematic overview of the delivery of CD46-Cas9 RNP complexes to primary human B cells by nucleofection. $2\times10^6$ primary human B cells were nucleofected with RNP complexes assembled with a two-component guide RNA (gRNA) protocol using tracrRNA, crRNA and recombinant *Streptococcus pyogenes* Cas9 nuclease. The cells recovered for 1 h after nucleofection before they were infected with wild-type (WT) EBV. The efficiency of gene editing was analyzed at the level of nucleotide sequence and surface protein. **(B)** Indel frequencies derived from next generation sequencing analysis of the second exon of the *CD46* gene are shown together with flow cytometry analysis of CD46 protein expression from all investigated samples. Cells were nucleofected with individual Cas9 RNP complexes (gRNA1 or 2) or concomitantly with both complexes and were analyzed 1 week later. Mean and standard deviation from four independent biological replicates are shown. **(C)** Flow cytometry analysis of CD46 surface staining of living cells treated with two RNP complexes 1 week post infection.

expression one week post nucleofection (**Fig 1B**) using the two specific gRNAs either individually or combined. When the CD46-Cas9 RNP complexes were used individually knockout efficiencies of about 60% were observed (**Fig 1B**). When both CD46-Cas9 RNP complexes were co-nucleofected knockout efficiencies regularly reached 85% and more both at the level of DNA editing and loss of CD46 surface expression (**Figs 1B and 1C** and **S1A**). The viability of nucleofected B cells typically ranged between 70 and 80% after nucleofection. The highest number of recovered cells was obtained with our optimized protocol as detailed in Materials and Methods. We found that minimizing the period of the cells in the electroporation buffer (P3 Primary Cell Nucleofector Solution), pre-cooling of the cuvette prior to nucleofection and supplementing the cells with prewarmed cell culture medium containing 20% fetal calf serum after nucleofection were the critical parameters. *CD46* knockout efficiencies showed little donor variations and nucleofection did not detectably interfere with subsequent EBV infection or outgrowth of lymphoblastoid cells.

A small fraction of peripheral blood mononuclear cells (PBMCs) consists of B lymphocytes, which are quiescent, resting naïve or memory B cells (**S2B Fig**). We purified B cells from PBMCs, nucleofected them with the two CD46-Cas9 RNP complexes and reached a knockout frequency of the *CD46* gene (**S2A–S2C Fig**) comparable to B cells prepared from adenoid tissue. We concluded that our protocol is equally applicable to resting human B cells from sources other than secondary lymphoid organs such as adenoids.

We also asked if we could use a standard protocol to expand the nucleofected B cells instead of infecting them with EBV. For this we cultivated primary B cells prepared from adenoid tissue on CD40 ligand feeder cells in the presence of IL-4 as described [20]. As expected this protocol also yielded *CD46* knockout cells with superior efficiency (**S2D Fig**) indicating that EBV infection ensures the cells' *in vitro* survival, only, but does not contribute to the efficiency of gene editing using the CRISPR-Cas9 technology.

## The kinetics of induced DNA breaks and their repair in non-infected and EBV infected primary B cells

To characterize the Cas9 mediated DNA cleavage and repair at the *CD46* locus further, we defined three different states of its exon 2. The 'intact' state is the original, unaltered DNA sequence that the CD46-Cas9 RNP complex targets. The 'indel' state characterizes the double-stranded DNA break (DSB) after its repair by the non-homologous end joining (NHEJ) pathway, which often introduces small insertions or deletions at the repaired site. Lastly, DNA repair may also lead to the 'deleted' state when a DNA fragment of about 98 nucleotides is lost upon the concomitant repair of two DSBs joining the distant DNA ends in exon 2 of the *CD46* locus.

To identify the kinetics of the different states, we isolated genomic cellular DNA from non-infected primary B cells at various time points after nucleofection. At each time point, we PCR-amplified a region that encompasses exon 2 of *CD46* with the two gRNAs target sites. To study the 'indel' and 'deleted' states, we set out to determine the activity of the repair machinery by measuring the accumulation of indels via next-generation sequencing of the PCR products. We achieved more than 70% indel frequency in primary human B cells 16 h after nucleofection and up to 85% 72h after nucleofection, before the non-infected residual B cells rapidly underwent apoptosis (**Fig 2A**). We also analyzed the PCR products on agarose gels and found a DNA fragment of approximately 180 bp in size already 16 hours after nucleofection indicative of the deleted state (**S1B Fig**). Together, these observations suggested a substantial DSB activity followed by an efficient DNA repair that caused indels and deletions in the majority of non-infected primary B cells within hours after nucleofection.

Next, we studied the kinetics of DSB and DNA repair in EBV infected cells in smaller temporal increments. The results showed a constant increase of knockout efficiencies that reached 60% 24 hours after nucleofection (**Fig 2B**). A couple of days later, knockout efficiencies plateaued at about 85%. In parallel, the loss of CD46 surface protein was studied by flow cytometry over time (**Fig 2C**). A small fraction of CD46 negative cells, about 10%, became detectable as early as 24 h after nucleofection. This fraction steadily increased to more than 85% on day 8, consistent with a moderate protein turnover of this surface receptor after the genetic knockout of *CD46*.

These results led us to conclude that Cas9 rapidly locates to its specific target site in resting primary human B cells and introduces DSBs with high efficiency. The endogenous DNA repair mechanisms are active in these primary cells independent of EBV-induced B cell activation (**Fig 2A**). Considerable fractions of DSBs are repaired already 16 hours after nucleofection as indicated by the deleted gene locus and the appearance of indels. The *CD46* knockout efficiency exceeded 85% when quantified at the protein level by flow cytometry one week after nucleofection which is in line with our genomic analyses.

## Nucleofection of the CD46-Cas9 RNP complex does not alter the overall B cells transcriptome

It has been reported that Cas9 remains attached to DNA ends after cleavage *in vitro* [21]. This roadblock together with a severed template will severely compromise transcription. In addition, the processes of nucleofection, DSBs induced by Cas9 and their repair might induce cellular stress that could also affect the transcriptome of the cells. To investigate possible adverse effects and to study global transcription as well as nascent CD46 transcripts we employed metabolic labeling of newly synthesized RNA with 4-thiouridine (4sU) (**Fig 2D**).

Isolated primary human B cells from three donors were nucleofected with the two CD46-Cas9 RNP complexes and infected with EBV one hour later. 23 hours after

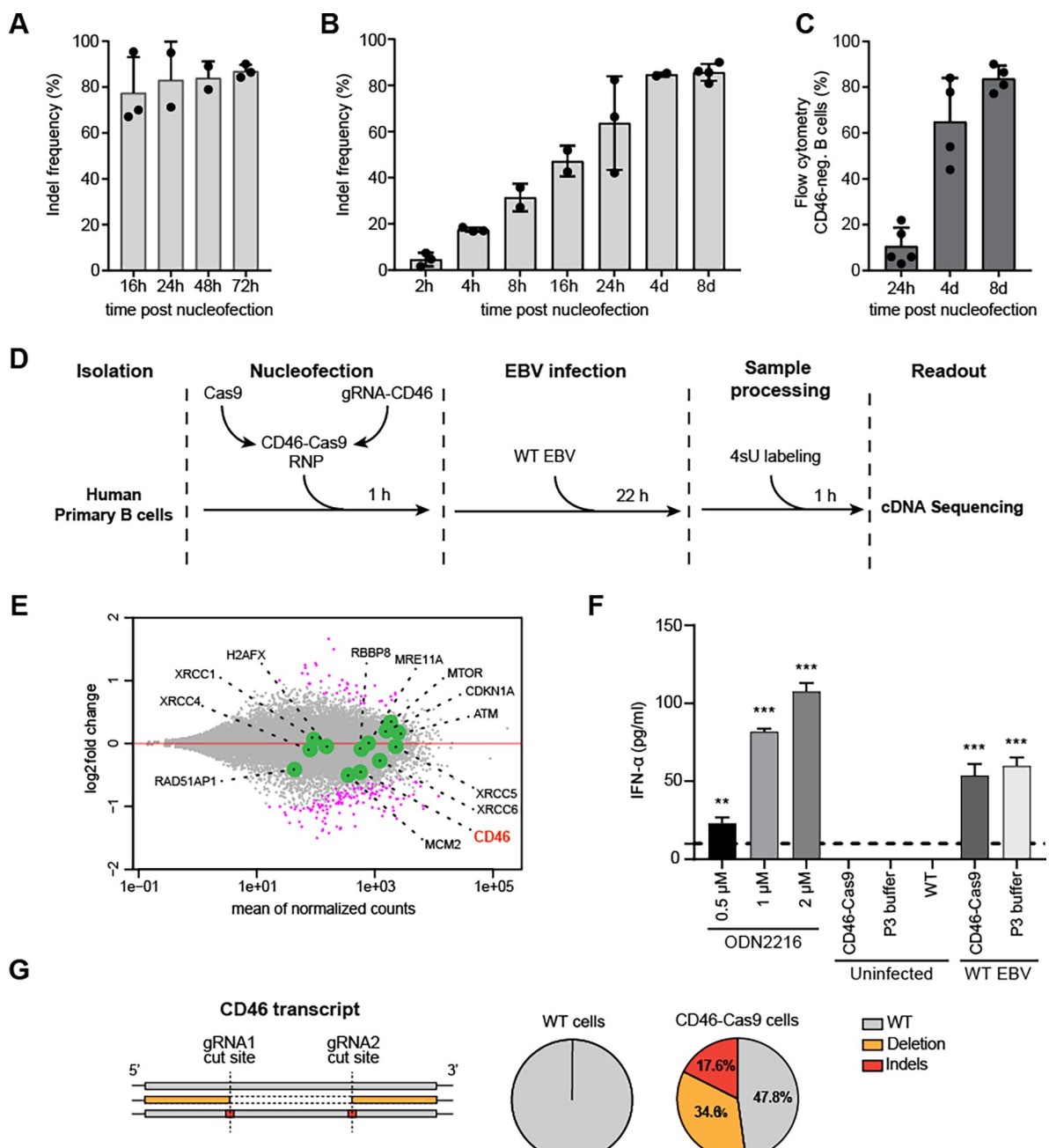

**Fig 2. The kinetics of induced DNA breaks and their repair in the second exon of *CD46* in non-infected and EBV-infected primary human B cells. (A)** Primary human B cells were nucleofected with the two CD46-Cas9 RNP complexes and analyzed 16, 24, 48 and 72 hours later. The cells were left non-infected. At the indicated time points viable cells were selected by Ficoll gradient centrifugation. Part of the *CD46* locus in their cellular DNA was amplified in a two level PCR barcoding scheme prior to next generation sequencing. Data were analyzed using the Outknocker 2.0 web tool. The results showed a indel frequency of 70% in primary human B cells 16 h after nucleofection in the absence of EBV infection. Indel frequencies increased up to 90% 72 h after nucleofection. Mean and standard deviation of independent biological replicates are shown. **(B)** The time course experiments show the indel frequencies in the second exon of *CD46* in EBV infected primary B cells after nucleofection with two CD46-Cas9 RNP complexes. Cellular DNA of viable cells was amplified in a two level PCR barcoding scheme prior to next generation sequencing and the data were analyzed with the Outknocker 2.0 web tool. Indel frequencies increased up to 85% 8 days after nucleofection. Mean and standard deviation of independent biological replicates are shown. **(C)** CD46 flow cytometry analyses of primary B cells 1, 4 and 8 days after nucleofection with two CD46-Cas9 RNP complexes and subsequent infection with WT EBV are shown. Mean and standard deviation of independent biological replicates are provided. **(D)** Scheme of metabolic labeling of newly transcribed RNAs with 4sU and their analysis by next generation sequencing. $2 \times 10^7$ primary human B cells were nucleofected with two CD46-Cas9 RNP complexes. Nucleofected (CD46-Cas9) as well as adjusted numbers of untreated (WT) B cell

samples were infected with wild-type (WT) EBV. 23 hours after nucleofection, newly transcribed RNAs were metabolically labeled with 4sU for 1 hour. After RNA extraction, biotinylation of newly transcribed 4sU labeled RNAs and their enrichment, cDNA libraries were established and sequenced on a NextSeq500 (Illumina) instrument with 2x150 bp paired end reads. 15.7 to 18.6 Mio reads per sample were obtained. **(E)** An MA plot shows the differentially expressed genes in CD46-Cas9 vs. WT B cell samples following EBV infection and 4sU labeling 24 hours after B cell preparation. Log2-fold changes and mean of normalized read counts were plotted on the y- and x-axes, respectively. 182 differentially expressed genes are designated by magenta dots. Green dots highlight the *CD46* gene and genes involved in the different pathways of DNA repair. Ongoing transcription at the *CD46* locus was intact in CD46-Cas9 RNP complex nucleofected cells but reduced by a factor of 0.73 24 h after nucleofection. **(F)** IFN-α release of primary B lymphocytes after RNP nucleofection. B cells nucleofected with the CD46-Cas9 RNP complexes or cells nucleofected with P3 buffer, only, were cultured with or without WT EBV infection overnight. On the next day, the cells were counted and re-seeded with identical cell numbers. After 48 hours, supernatants were collected and IFN-α was measured by ELISA. As positive controls, uninfected cells were treated with different concentration of the TLR9 agonist ODN2216 for 20 hours prior to analysis. The threshold level of detection was 10 pg/ml IFN-α as indicated by the dashed line. P values were calculated using the one-way ANOVA test. ***, $P < 0.001$, **, $P < 0.01$. Mean and standard deviation of two biological and technical replicates are shown. **(G)** Shown are three schematic examples of mapped reads aligned to the hg19 reference sequence together with the two RNP complex target sites (chr1: 207,930,419–207,930,438 and chr1: 207,930,497–207,930,516) at exon 2 of the *CD46* gene. Reads with unmodified exon 2 (WT) sequences, reads with nucleotide mutations (indels, i.e. base changes or small insertions and deletions), reads with deletions in between the two annotated RNP complex target sites and their percentages are shown.

nucleofection newly transcribed RNAs were metabolically labeled with 4sU for 60 minutes. Non-nucleofected (WT) but equally EBV infected and 4sU labeled cells served as controls. The labeled RNA was thiol-specifically biotinylated and separated from unlabeled, preexisting RNA. After validation and quantification of the labeled and enriched RNAs cDNA libraries were established to perform transcriptome sequencing. The overall RNA yield was very low but a thorough quality control after RNA extraction and library preparation confirmed the high quality of all six libraries.

The transcriptomes of CD46-Cas9 and WT B cell samples were analyzed and the comparison was visualized in an MA plot (**Fig 2E**). After normalization, 182 transcribed genes were found to be significantly regulated in this comparison (**S1 Table**). In nucleofected cells transcription at the *CD46* locus was reduced by a factor of 0.73 as compared to control cells, but the locus was clearly transcribed already 24 hours after CD46-Cas9 RNP complex nucleofection and was not blocked by, e.g., the cellular DNA repair machinery (**Fig 3A**). In the MA plot we also highlighted transcripts of genes involved in the different pathways of DNA repair. No clear trend emerged from this analysis suggesting that nucleofection of CD46-Cas9 RNP complexes has no major impact on the transcriptome of human B cells. Gene ontology (GO) and gene set enrichment analysis (GSEA) using Webgestalt (http://www.webgestalt.org/) provided lists of gene sets that were differentially regulated such as 'nucleosome' and 'DNA packaging complex' among others that seemed to be less conclusive to us.

Using the transcriptomic data of the *CD46* gene we analyzed the occurrence of mutations in exon 2 of the CD46 RNA transcript. Reads were inspected using the IGV browser and were assigned to three different groups: reads with unmodified exon 2 sequences, reads with 98 nucleotide deletions (Deletion) and reads with base changes or small deletions (Indels) (**Fig 2G**). All reads from untreated (WT) B cell samples were identical to the reference sequence, whereas only 47.8% of the reads in CD46-Cas9 nucleofected RNA samples were unaltered. 17.6% of the reads in these RNA libraries showed mutations whereas the remaining 34.6% documented the deletion of about 98 nucleotides as expected 24 h after nucleofection (**Fig 2G**).

We had an even closer look at exon 2 of the transcribed CD46 gene and its read profiles in non-nucleofected, untreated cells and in cells 24 hours after CD46-Cas9 RNP complex nucleofection (**Fig 3A and 3B**). Apparently, the overall read coverage of exon 2 RNA was reduced to about one third in nucleofected cells compared with RNA data from non-nucleofected B cell samples. The substantial differences in reads were specific of exon 2 of *CD46*, the target sites of the two gRNAs whereas exons of other transcripts of equal length (189 nucleotides) did not

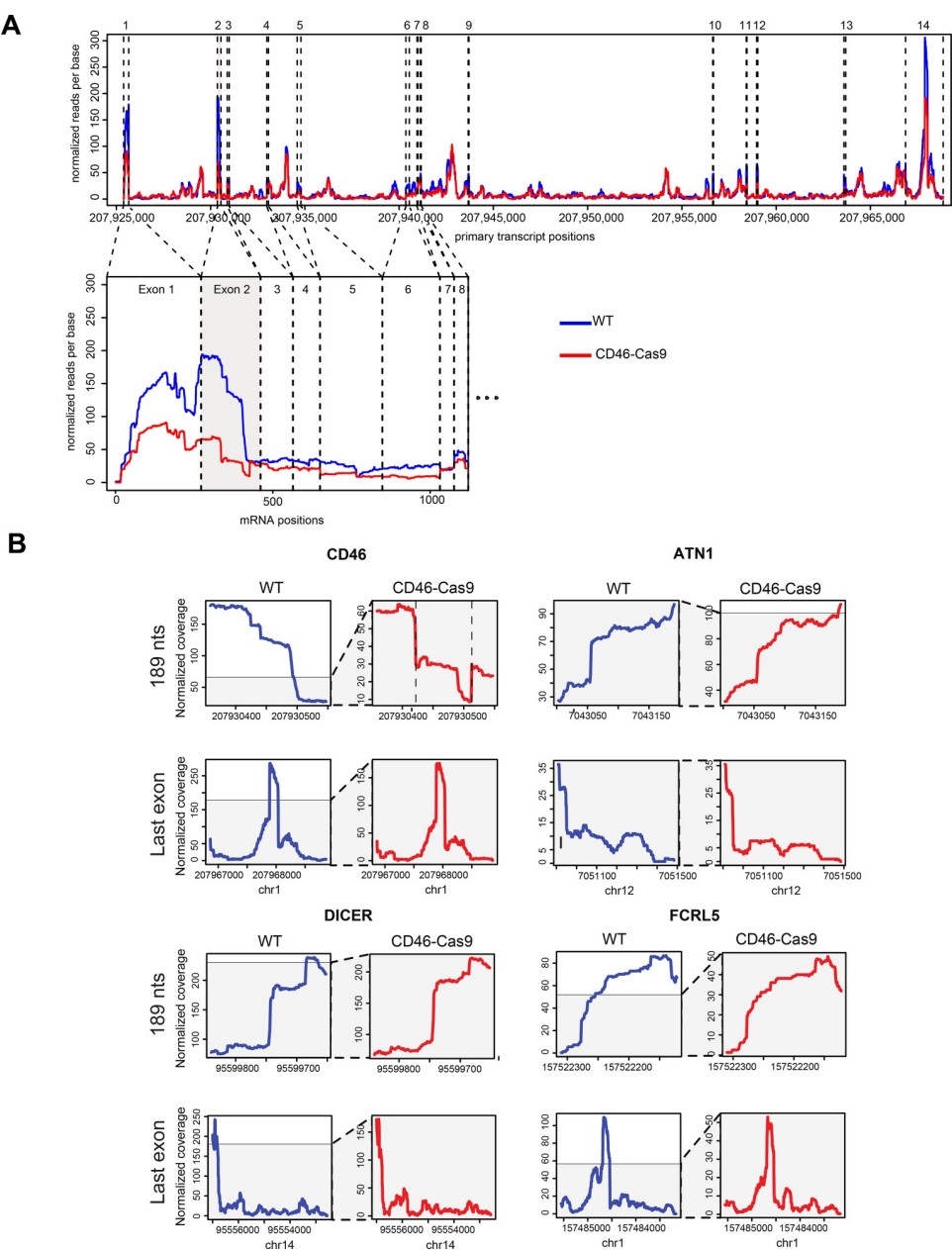

**Fig 3. Read coverage of 4sU labeled transcripts shows a reduced transcription in cells 24 hours after CD46-Cas9 RNP complex nucleofection only at exon 1 and 2 of CD46. (A)** The upper plot shows the primary, unspliced transcript of the CD46 gene together with its nucleotide coordinates on chromosome 1 and the 14 exons (dashed, vertical lines) of the CD46 transcript (x-axis). The normalized read coverage at single nucleotide resolution is shown (y-axis) from WT cells (blue line) and CD46-Cas9 treated cells (red line). The lower graph depicts the mRNA data of exon 1 to 8 of the CD46 transcript covering the two conditions. Read normalization of the three pairs of replicates was performed as stated in Materials and Methods. **(B)** The plots show the primary transcript positions (bp) (x-axis) vs. the normalized read coverage at single nucleotide resolution of four selected transcripts (CD46, ATN1, DICER, FCRL5). Transcript data (y-axis) from WT cells (blue line) and CD46-Cas9 nucleofected cells (red line) are shown. Plots indicated '189 nts' visualize the 189 nt long exons of the ATN1, DICER and FCRL5 transcripts together with the second exon of the CD46 transcript. Plots of the last exons of the four transcripts are shown for comparison. The two dashed vertical lines in the plot of the second exon of CD46 indicate the positions of the Cas9 mediated cleavage sites.

show similar differences (**Fig 3B, upper panels**). Similarly, profiles of the last exon of CD46 RNA compared with the last exons of other transcripts were comparable (**Fig 3B, lower panels**).

In addition to the detailed transcriptomic analyses, we asked whether nucleofection of Cas9-gRNA ribonucleoprotein complexes (probably together with free, non-complexed tracrRNA and crRNA) triggers a type I interferon (IFN) response indicative of innate immunity of the manipulated B cells although our analysis in **Fig 2E** and data in **S1 Table** did not point into this direction. To test this, we nucleofected B cells purified from adenoid tissue with CD46-Cas9 RNP complexes and measured the concentration of IFN-α in the culture medium. IFN-α release in the supernatants of nucleofected cells was undetectable and thus below the threshold level of detection in our assay (**Fig 2F**). Only EBV infected cells released low levels of IFN-α as expected in Fig 2F and [22]. Our analysis indicates that RNP complex delivery to primary human B cells via electroporation does not induce a possibly adverse type I interferon response in these cells.

We can also conclude that Cas9 nucleofection and local DNA breaks have no global impact on ongoing cellular transcription. The experiments document that more than 50% of exon 2 mRNA transcripts of *CD46* contain indels or are deleted (**Fig 2G**) indicating that cleavage and repair of template DNA are already completed and transcription of the edited gene has resumed and is functional as early as 24 hours post nucleofection.

## EBNA3C controls *CDKN2A*, a barrier to B cell proliferation

Next, we explored whether editing of a specific gene in primary human B cells with well-established functions during EBV infection would result in a phenotype. We targeted *CDKN2A*, which encodes the p16$^{INK4a}$ protein, a CDK4 inhibitor that controls cell cycle G1 progression. This gene, which is as an important tumor suppressor gene is often lost or mutated in various tumor types. The work by Martin Allday's group suggested that EBNA3C, a transcriptional repressor and latent EBV gene induces the epigenetic silencing of *CDKN2A* in latently infected lymphoblastoid cell lines [16]. Repression of *CDKN2A* was reported to be a prerequisite to support cell proliferation long-term as shown in lymphoblastoid B cell lines infected with an EBV derivative with a conditional EBNA3C allele [16]. A single infection experiment with primary B cells from an individual with a genetic lesion that prevents expression of functional p16$^{INK4a}$ is the only direct evidence supporting this claim in a physiological context [16].

We sought to validate the role of *CDKN2A* in EBV infection using our CRISPR-Cas9 approach by inactivating this gene in primary human B cells from different donors. We designed a gRNA that targets exon1α of the *CDKN2A* locus. As shown schematically in **Fig 4A** Cas9 introduced knockouts in this exon will cause frame shift mutations affecting the translation of p16$^{INK4a}$. To this end, control (WT) and nucleofected (p16 KO) cells were infected with the WT (6008) EBV strain or the ΔEBNA3C (6123) mutant EBV [14] to establish four different lymphoblastoid cell lines (LCLs) from each donor. After one week, the *CDKN2A* gene showed a knockout efficiency of about 60 to 80% as expected from a single DSB that occurred in all cells (**S3 Fig**).

Next, we wanted to study the biological consequences of the *CDKN2A* knockout. Primary WT B cells and p16 KO cells were mixed such that the latter comprised 10 to 20% of the total population, only, at the start of the experiments. We infected these mixed cell populations with either the WT EBV strain or the ΔEBNA3C mutant EBV and cultivated the infected cells for up to 10 weeks. Occasionally, cells were sampled and their cellular DNAs were analyzed for *CDKN2A* knockout frequencies by next generation sequencing. As shown in **Figs 4B** and **S4A**, cells infected with the WT EBV strain showed a low fraction of inactivated *CDKN2A* genes in

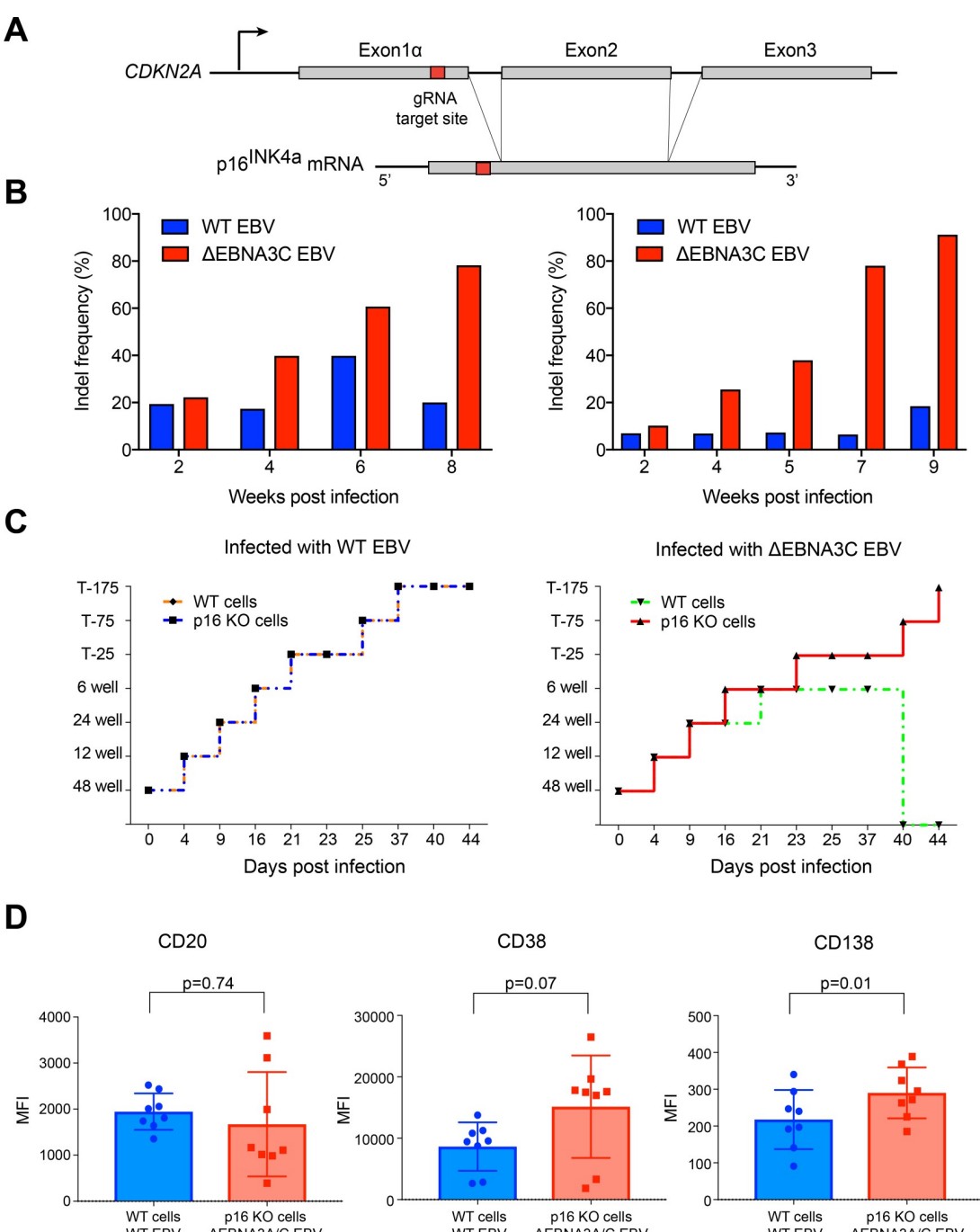

**Fig 4. p16^{INK4a} is a functional barrier to EBV driven proliferation of lymphoblastoid cells. (A)** Blueprint of the primary transcript and the spliced mRNA with the three exons of *CDKN2A* on chromosome 9 encoding the p16^{INK4a} protein. The target site of the RNP complex within the 1st exon (exon1α) (chr9:21,974,678–21,974,827) is shown. **(B)** Study of the biological effect of the *CDKN2A* knockout in a time course experiment. WT and p16 KO cells were mixed such that the fraction of the latter was in the order of 10 to 20%, when the cells were infected with WT or ΔEBNA3C EBV strains. The knockout status of the *CDKN2A* gene was studied by next generation sequencing to analyze the CD46 locus of the mixed cell populations over time. The fraction of cells with a disabled *CDKN2A* gene increased in cells infected with ΔEBNA3C EBV exceeding 80% after eight weeks, whereas the knockout status of *CDKN2A* in the population of cells infected with WT EBV did not show a clear trend. Results from two biological replicates are shown, additional replicates can be found in **S4A Fig**. **(C)** Cell numbers of four different B cell populations were plotted as a function of days post nucleofection (x-axis) versus the format of the cell culture vessel (y-axis) starting with a single well in a 48-well cluster plate. 2×10^6 B cells with an intact *CDKN2A* locus (WT cells) or cells with an edited *CDKN2A* gene (p16 KO cells) were infected with wild-type (WT) EBV (left panel) or ΔEBNA3C EBV (right panel). WT EBV

infected primary human B cells developed into stably expanding lymphoblastoid cell lines irrespective of their p16$^{INK4a}$ status (left panel). ΔEBNA3C EBV infected B cells could be expanded until about six weeks p.i., when cells with an intact *CDKN2A* gene ceased to proliferate and were lost eventually (right panel). In contrast, ΔEBNA3C mutant EBV infected p16$^{INK4a}$ negative B cells continued to proliferate beyond this time point. The results were consistent between four different biological replicates. Other replicates are shown in **S4B Fig.** (D) p16 KO cells infected with ΔEBNA3A/C EBV and WT cells infected with WT EBV were analyzed for CD20, CD38 and CD138 cell surface levels by flow cytometry. The graph summarizes the mean fluorescence intensities (MFI) of CD20, CD38 and CD138 markers of living cells two months post infection. Mean and standard deviation obtained from 8 independent biological replicates are shown. The significance of MFI values of the immunophenotypic surface proteins was calculated using the Wilcoxon matched-pairs signed rank test.

the population of cells in the time course of the experiments. In contrast, cells infected with the ΔEBNA3C mutant EBV showed a slow but gradual increase of *CDKN2A* indel frequencies that reached and exceeded 80% in some experiments (**Figs 4B and S4A**). This observation documented a selection of *CDKN2A*-negative cells over time, when the cells were initially infected with an ΔEBNA3C mutant EBV. In the absence of EBNA3C, cells expressing p16$^{INK4a}$ are lost because they enter a G1 block and thus cease to proliferate as suggested previously, for a recent review see [23]. On the contrary, cells that lack p16$^{INK4a}$ and are infected with the ΔEBNA3C mutant EBV have a survival advantage and continue to proliferate even in the absence of the viral transcriptional repressor EBNA3C (**Figs 4B and S4A and S4B**), which leads to their enrichment in the mixed cell population.

We also observed the proliferation of the four different cell cultures by simply tracking their cellular expansion. We started with 2x10$^6$ primary B cells infected with the WT EBV strain or the ΔEBNA3C mutant EBV in a single well of a 48-well plate and continued to expand the cultures gradually to a final volume of about 100 ml in T175 cell culture flasks. WT EBV infected primary human B cells developed to LCLs irrespective of their p16$^{INK4a}$ status and reached the final culture volume about 5 to 7 weeks after the start of the experiments (**Figs 4C, (left panel) and S4C**). ΔEBNA3C mutant EBV infected B cells could be expanded similarly until about 5 to 6 weeks p.i., when p16$^{INK4a}$ positive B cells ceased to proliferate and eventually died (**Figs 4C, (right panel) and S4C**). In contrast, p16$^{INK4a}$ negative B cells infected with the ΔEBNA3C mutant EBV strain continued to proliferate beyond this time point.

Together, our experiments with primary B cells confirmed that EBNA3C is essential for the repression of *CDKN2A*, a finding which is in line with reports of Martin Allday's group. All results are consistent with p16$^{INK4a}$ being the functional barrier to EBV-driven proliferation of LCLs. EBNA3C plays a central function to restrain transcription of p16$^{INK4a}$ in these cells as suggested previously [23–25].

## p16$^{INK4a}$ negative B cells infected with an EBV strain devoid of EBNA3A and C do not readily differentiate into the plasma cell lineage

p16$^{INK4a}$ is an obstacle to establish EBNA3C deficient LCLs, whereas its cousin, EBNA3A is dispensable [26,27]. Both viral proteins have been implicated in blocking B cell differentiation towards plasmablasts and plasma cells [17], because they can act as epigenetic repressors of several cellular loci other than *CDKN2A* such as *CDKN2C* and *PRDM1*, which encode p18$^{INK4c}$ and BLIMP-1, respectively, and thus prevent terminal differentiation of lymphoblastoid B cells [17]. Using our p16$^{INK4a}$ knockout strategy, we revisited this finding with the aim to obtain human plasma cells in large quantities by removing the p16$^{INK4a}$-mediated block of cellular proliferation. Towards this end, we infected mature B cells with an ΔEBNA3A/C (6331) double knockout EBV [14] after nucleofection with the *CDKN2A* specific Cas9 complex. We expanded the cultures and analyzed the differentiation status of the infected B cells by flow cytometry to assess their phenotypes using diagnostic and paradigmatic cell surface

markers [28]. B cells from the same donors infected with the WT EBV strain served as controls (**Fig 4D**).

The immunophenotypic characteristics of plasma cells include surface expression of CD138, elevated levels of CD38, a general marker of lymphocytic activation, and down-regulation of CD20 [28]. The multiple myeloma cell line RPMI-8226, which expresses high levels of CD138 and contains a typical small subpopulation of CD138high cells (**S5 Fig**), as reported previously [29,30], served as positive control.

We prepared and infected primary B cells as described above, cultivated them for up to two months and evaluated their immunophenotypic surface pattern. We found that cells infected with the two viruses did not show substantially different levels of the three surface markers CD20, CD38 and CD138 (**Fig 4D**). The levels were either not significantly different in case of CD20 and CD38 or significantly different but very low in case of CD138 and without a discernable CD138high subpopulation as seen in RPMI-8226 cells (**S5 Fig**) even though ΔEBNA3A/C mutant EBV infected cells had a trend towards lower levels of CD20 and higher levels of CD38 and CD138 as compared to WT EBV infected cells. Although this trend is concordant with immunophenotypic characteristics of plasma cells, our results with newly infected primary B cells appear to differ from previous work in which EBNA3A/C was found to block plasmablasts or plasma cell differentiation in established LCLs [17].

## Discussion

In the current study, we established nucleofection-based delivery of Cas9 RNP complexes targeting *CD46* as a model gene with knockout efficiencies of 85% and beyond in polyclonal, quiescent primary human B cells. Our readily applicable protocol, based on commercial reagents, allows genome-wide editing in primary resting human B cells.

Having worked with primary human B cells for decades, our laboratory has been unsuccessful in genetically manipulating primary human B cells very much in contrast to established B cell lines [31,32]. Using the CRISPR-Cas9 approach, primary human B cells from many donors tolerated Cas9 RNP complex nucleofection with 70 to 80% surviving cells, which is in stark contrast to transfections with DNA [11].

The ease and efficiency of genetic manipulation of primary B cells with our protocol is a considerable step forward following pioneering work in primary human T cells [4–6,33]. The high *CD46* knockout efficiency in primary human B cells likely resulted from combinatorial effects of two CD46-Cas9 RNP complexes and optimized nucleofection conditions. Conceivably, the specific gene locus and the design of the gRNAs may also strongly affect the outcome. We followed the scheme proposed by Clarke et al. [34] and designed the upstream and downstream CD46-Cas9 RNP complexes to target the non-template and template orientations of the upper and lower DNA strands, respectively. Remarkably, this strategy also induced an efficient deletion together with frame shift mutations that led to a functional *CD46* knockout exceeding 85% at the protein level (**Fig 1B and 1C** and **Fig 2A, 2B and 2C**). Similarly, when we targeted the *CDKN2A* gene with a single Cas9 RNP complex we regularly observed frame shift mutations in the order of 75% (**S3 Fig**) suggesting that nearly all cells in the B cell population were nucleofected initially.

Clearly, CRISPR-Cas9 mediated gene editing did not depend on subsequent infection with EBV (**Fig 2A**) indicating that EBV does not promote or enable gene editing but is only the driver of initial B cell survival and proliferation, which starts four days post infection [14]. Importantly, gene editing was similarly efficient with resting B cells isolated from peripheral blood (**S2 Fig**) documenting that our protocol does not depend on an preactivated cellular state of the B cells. Of note, a previous report concluded that activated human B cells are much

more amenable to Cas9-mediated gene editing compared to resting B cells with only about 20% knockout efficiency targeting a single gene during quiescence [8]. Delivery of the RNP complex by electroporation did not shown any effect on cellular transcription (**Fig 2E**), which might arise when lentiviral vectors are used for cellular delivery of gRNAs [35]. Together, our data demonstrate that Cas9-mediated gene editing is fully functional and efficient in resting primary human B cells.

Both the overall efficiency and the kinetics of gene editing in resting B cells using our protocol were rather unexpected. Already 24 hours after Cas9 RNP complex nucleofection more than 50% of newly transcribed CD46 mRNAs contained indels or showed the characteristic deletion in exon 2 (**Fig 2G**) indicating that DSB, their subsequent DNA repair and clearance of the templates to allow transcription can occur within hours following delivery of the Cas9 RNP complexes. The maximum knockout efficiency in the *CD46* locus typically exceeded 85% one week after nucleofection, indicating that the CD46-Cas9 RNP complex remains active for several days in these cells, contrary to a previous report in which the authors used established human cells [36].

Focusing on genetic manipulation of *CDKN2A*, we employed our novel protocol and studied two related facets of EBV's complex biology in primary human B cells. EBNA3A and C, two related latent EBV genes, have been reported to act as mediators of epigenetic repression of several cellular gene loci. Among them is *CDKN2A*, which encodes the cell cycle inhibitor p16$^{INK4a}$ and which reportedly blocked proliferation of latently EBV-infected B cells in the absence of EBNA3C's repressive epigenetic functions [23–25]. Our results after *CDKN2A* knockout shown in **Fig 4** corroborated this notion with primary B cells from six donors recapitulating a situation with B cells from a rare p16$^{INK4a}$ deficient individual [16].

We went on and investigated another biological function suggested for EBNA3A and EBNA3C. A seminal paper by the Allday laboratory reported that EBV-infected lymphoblastoid B cells differentiated spontaneously and acquired a plasma cell-like phenotype in the absence of EBNA3A and C using conditional EBNA3 alleles [17]. The authors also discussed that inactive EBNA3A and C alleles may permit EBV infected B cells to differentiate further into plasmablasts and fully differentiated functional plasma cells, eventually, solely upon silencing of p16$^{INK4a}$. Since authentic human antibodies have a promising translation potential, we tested this interesting hypothesis. It implies that *CDKN2A* indirectly precludes the differentiation of cycling pre-plasmablasts [17] because p16$^{INK4a}$ acts as a roadblock of the cell cycle preventing their proliferation and subsequent differentiation [37]. We genetically tested this scenario but found no clear evidence that EBNA3A and C together block differentiation of lymphoblasts to pre-plasmablasts, plasmablasts or plasma cells (**Figs 4D and S5**). This suggests that EBV-infected and activated B cells do not follow the default pathway to differentiate into plasma cells spontaneously *in vitro* for so far unknown reasons.

The CRISPR-Cas9 technology has gained much attention and one of the first phase I clinical trials assesses the feasibility of CRISPR-engineered T cells in patients with refractory cancer [38]. B cell therapies have a similar potential but the obstacle to engineer primary human B cells has prevented translational progress with autologous B cells in patients. Our newly developed CRISPR-Cas9 protocol can be successfully applied to primary human B cells and will now also enable studies into the functional interactions between EBV and its primary host cells.

## Methods

### Ethics statement

PBMCs in the form of buffy coats from healthy adult donors were purchased from the Institute for Transfusion Medicine, University of Ulm, Germany. The buffy coats were a side blood

product unsuited for clinical use and were delivered to us in fully anonymized form. Where indicated, primary naive B lymphocytes were isolated from adenoid samples. They were left-overs from adenoidectomies performed at the Department of Otorhinolaryngology and Head and Neck Surgery, Klinikum der Universität München; these samples also were transferred to our institution in fully anonymized form. The institutional review board (Ethikkommission) of the Klinikum der Universität München, Munich, Germany) approved this procedure (project no. 071-06-075-06).

## Eukaryotic cell lines

RPMI-8226 cells derived from a multiple myeloma biopsy [39,40] and Raji cells [41] were purchased from German Collection of Microorganisms and Cell Cultures GmbH (DSMZ), Braunschweig. RPMI-8226 cells, Raji cells and LCLs (derived from EBV infected B cells) were cultivated in RPMI-1640 medium supplemented with 8% FCS, 100 μg/ml streptomycin, 100 U/ml penicillin, 1 mM sodium pyruvate, 100 nM sodium selenite, and 0.43% α-thioglycerols at 37˚C and 5% $CO_2$. HEK293 based EBV producer cell lines [14] were cultured in the same medium with 500 ng/ml puromycin.

## Virus supernatant production and quantification

To obtain virus stocks, three different HEK293 producer cell lines established in our lab (6008, 6331, 6123) [14] were transiently transfected with expression plasmids encoding BZLF1 (p509) and BALF4 (p2670) to induce EBV's lytic cycle as described previously [14,42]. Briefly, cells were seeded onto 13 cm dishes to reach 80% confluency. The next day, the cells were transfected using 6 μg p509 and 6 μg p2670 plasmid DNAs [42] mixed with 72 μl of a 1 mg/ml stock solution of polyethylenimine (#24765; Polysciences) in water (pH 7.0) in 2.4 ml plain RPMI1640 cell culture medium and incubated for 15 min. The mix was dropped onto HEK293 producer cell lines in fully supplemented cell culture medium without puromycin and the transfected cells were incubated for 3 days. Cell culture supernatants were collected and centrifuged for 8 min at 300 g and then at 1,200 g for 8 min. To titer the virus stocks, $1x10^5$ Raji cells (obtained from the Leibniz Institute DSMZ) were incubated with different volumes of virus stocks at 37˚C for 3 days. The percentages of GFP-positive cells were determined by flow cytometry using a BD FACS Canto instrument (BD Bioscience), and the linear regression equation was calculated as previously described (Steinbrück et al., 2015). The virus stocks were stored at 4˚C.

## B cell isolation from adenoid tissue

Primary B cells were isolated from adenoid biopsies of anonymous donors. The adenoid tissues were rinsed with PBS several times using a 100 μm cell strainer (Falcon), transferred to a sterile petri dish and mechanically chopped with two sterile scalpels and PBS. The cell suspension was filtered through a 100 μm strainer. This procedure was repeated several times to recover a maximum number of cells. The volume of the collected cells was increased to 30 ml with PBS. 0.5 ml defibrinated sheep blood (Thermo Scientific Oxoid, catalog no. SR0051D) was added to deplete T cells in the next step. The cell suspension was underlaid with 15 ml Ficoll Hypaque and the samples were centrifuged at 500 g for 30 min. Cells were carefully collected from the interphase and transferred to a new 50 ml tube. The cell suspension was washed three times in a total volume of 50 ml PBS and centrifuged with decreasing sedimentation forces (450, 400 and 300 g) for 10 min each. The cell pellet was resuspended in pre-warmed complete cell culture medium supplemented with Ciprobay (1:200 dilution) to prevent bacterial contamination.

## B cell isolation from PBMCs

B cells were isolated from fresh LRSCs (Leukoreduction system chambers) after plateletpheresis procedures. The blood cell sample was diluted 1:6 or more in PBS and underlaid with 15 ml Ficoll Hypaque in a 50 ml Falcon tube. The samples were centrifuged at 500 g for 30 min. PBMCs were carefully collected from the interphase and transferred to a new 50 ml tube. The cell suspension was washed three times in a total volume of 50 ml PBS and centrifuged with decreasing sedimentation forces (450, 400 and 300 g) for 10 min each. The cell pellet was resuspended in fresh medium and B cells were isolated using the B cell isolation kit II (# 130-091-151; Miltenyi Biotec) as described in manufacturer's protocol using MACS sorting with LS column.

## B cell activation using CD40 ligand feeder cells together with IL-4

B lymphoblasts were generated by plating B cells isolated from adenoids on irradiated CD40 ligand feeder cells in the presence of 2 ng/ml IL-4 as described by Wiesner et al. [20].

## ELISA

For ELISA assays, CD46-Cas9 RNP complex or P3 buffer nucleofected B cells were cultured with or without EBV infection at 37˚C for 20 h. On the next day, the cells were counted and $2 \times 10^5$ cells in 200 µl total cell culture were plated in 96 well cluster plates. After 48 h, cell supernatants were collected and IFN-α secretion was quantified using the IFN-α pan Human ELISA development Kit (#3425-1A-20; Mabtech).

## Infection of primary human B cells with EBV

Virus supernatants were incubated with primary B cell suspension at an optimal multiplicity of infection (MOI) of 0.1 'green Raji units' overnight as described [43]. The following day, the virus supernatant was replaced with fresh medium after centrifugation of the cell suspension at 300 g for 10 min.

## Staining of B cell surface markers for flow cytometry

B cells were collected at different time points after infection and stained with antibodies specific for the indicated cell surface markers; CD19 (efluor450 conjugated, eBioscience, #48-0199-42, diluted 1:50), CD20 (PE-conjugated, BD, #556633, diluted 1:100), CD38 (PECy7-conjugated, eBioscience, #25-0389-42, diluted 1:300), CD138 (APC-conjugated, Invitrogen, #17-1389-42, diluted 1:50) and CD46 (APC-conjugated, Biolegend, #352405, diluted 1:50), CD27 (APC-conjugated, Biolegend, #302810, diluted 1:100) and IgD (FITC-conjugated, BD, #555778, diluted 1:100). For staining, the cells were washed with and resuspended in 50 µl FACS staining buffer (PBS, 0.5% BSA, 2 mM EDTA). After adding the antibodies, samples were mixed and incubated at 4˚C in the dark for 20 min. Cells were washed with 1 ml of FACS staining buffer and resuspended in 200 µl FACS staining buffer. Flow cytometry data were collected on a FACS Canto II (BD) or FACS Fortessa (BD) instrument. The FACS data were analyzed with the FlowJo software (version 10.4).

## Assembly of ribonucleoprotein (RNP) complexes

Chemically synthesized tracrRNA and crRNAs were obtained from IDT and were dissolved in Nuclease-free Duplex Buffer (IDT) at 200 µM. 5 µl of crRNA and tracrRNA each were mixed 1:1 to reach a final concentration of 100 µM. To form the gRNA complex the mixture was incubated at 95˚C for 5 min in a pre-heated PCR cycler and slowly cooled down to room

temperature by switching the instrument off. 6.5 μl of a 62 μM Cas9 V3 protein preparation (1081059, Integrated DNA Technologies) was added to 10 μl of the annealed gRNAs and the mix was diluted with PBS to a final volume of 50 μl. Prior to adding PBS, it was filtered (0.22 μm mesh size) and used at room temperature. The mixture was incubated at room temperature for 10 min to form RNPs at a final concentration of 8 μM Cas9. The gRNAs specific for the target gene were designed using the IDT web site. The sequences of the crRNAs are provided in **S2 Table**.

## Nucleofection of RNP complexes into primary human B cells

$2×10^6$ freshly isolated primary B cells were washed in PBS and resuspended in 20 μl P3 Primary Cell Nucleofector Solution buffer prepared with Supplement 1 buffer (Lonza) according to the manufacturer's instructions (P3 Primary Cell 4D-Nucleofector X Kit S). Cells were mixed with 5 μl of the RNP mixture by gently pipetting and were transferred to pre-cooled (4°C) 16 well Nucleocuvette Strips (Lonza). Primary human B cells were nucleofected using the EH100 program of Lonza´s protocol. 100 μl prewarmed non-supplemented RPMI1640 medium was added to the cells, which were incubated for 15 min at 37°C. The cells were transferred to a single well of a 24-well cluster plate and complete prewarmed cell culture medium containing 20% FCS was added to a final volume of 220 μl to allow cell recovery. The cells were incubated at 37°C, 5% $CO_2$ for 1 h before they were infected with EBV.

## Knockout quantification with next generation sequencing

Genomic DNA was isolated from $0.5–1×10^5$ cells after Ficoll Hypaque density gradient centrifugation. The cell suspension was underlayered with 5 ml Ficoll Hypaque in a 15 ml tube and the sample was centrifuged at 500 g for 30 min. Cells were carefully collected from interphase and transferred to a new 15 ml tube. The cell suspension was washed with PBS 2 times in a total volume of 15 ml (450, 400 g) for 10 min each. The last wash was performed in a total volume of 1 ml (300 g) for 10 min. The cells were resuspended in 20 μl of lysis buffer (0.2 mg/ml proteinase K, 1 mM $CaCl_2$, 3 mM $MgCl_2$, 1 mM EDTA, 1% Triton X 100, 10 mM Tris pH 7.5) supplemented with 20 ng/ml proteinase K. The reactions were incubated at 65°C for 10 min and at 95°C for 15 min in a PCR thermo cycler. In a two-step PCR barcoding scheme, first-level PCR reactions were performed using 5 μl of the cell lysate in a final total volume of 25 μl that included primers, dNTPs, reaction buffer, GC-enhancer buffer and 0.25 μl of Q5 Hot Start High-Fidelity DNA Polymerase (NEB, Massachusetts, USA). Amplification was done after an initial phase of denaturation (95°C, 5 min) in 32 cycles composed of 98°C, 62°C, and 72°C for denaturation, annealing and elongation and for 20 sec, 30 sec, and 40 sec, respectively. After an analytical gel, 6 μl were transferred to a second round of PCR amplification using barcode primers that were sample specific for subsequent DNA sequencing. The PCR reaction was run on a 1.5% preparative agarose gel, size-separated and bands of the expected lengths were eluted and quantified using a Nanodrop spectrophotometer system (Thermo Fisher). Next generation sequencing was performed using a MiSeq benchtop sequencing system (Illumina). Data were obtained in FASTQ format and analyzed using the Outknocker 2.0 web tool (http://www.outknocker.org/outknocker2.htm) [44]. Sequences of all primers are provided in **S3 Table**.

## Metabolic labeling of newly transcribed RNA with 4sU and RNA isolation

$2×10^7$ primary human B cells were nucleofected with the CD46-Cas9 RNP complex as described in the section entitled 'Nucleofection of RNP complexes into primary human B cells. CD46-Cas9 B cell samples as well as the adjusted numbers of non-nucleofected B cell samples

were infected with the wild-type EBV. Previously published 4sU protocols [45] were adjusted to our protocol as follows. Newly transcribed RNAs were metabolically labeled with 4sU (Sigma Aldrich, T4506-25mg) by adding 4sU to the culture medium at a final concentration of 100–200 μM and incubation at 37°C for 1 hour. At the end of labeling, the cells were centrifuged and washed with cold PBS. For RNA extraction, the cell pellet was homogenized in 1 ml of 4°C cold Trizol reagent by vortexing and passing the lysate through a 25G needle 8–10 times. The samples were vigorously vortexed for 15 sec and incubated for 2–3 min at room temperature. The lysate was transferred into a prepared 5PRIME Phase Lock Gel–Heavy and Light (Quantabio) and centrifuged at 12,000 g for 15 min at 4°C to separate RNA, DNA and proteins into different phases. The upper phase containing RNA was carefully transferred to a new sterile RNase-free tube and combined with an equal volume of isopropanol and 0.1 volume of 5 M NaCl. The mixture was incubated for 10 min at RT and precipitated RNA was collected at 12,000 g at 4°C for 10 min. The pellet was washed with 1 ml of 80% of ethanol and spun at 5,000 g for 5 min at 4°C and air-dried for 2–3 min at room temperature. The pelleted RNA was rehydrated in 20–40 μl of RNase-free water.

To biotinylate the newly transcribed and 4sU labeled RNAs, 1 μg/μl of total RNA was incubated with biotin-HPDP (Pierce, 1 mg/ml) in biotinylation buffer (1 M Tris ph 7.4, 0.5 M EDTA) while agitating in the dark for at least 2 h. The identical volume of phenol-chloroform-isoamyl alcohol was mixed with the biotinlyated RNA sample, which was later purified by using Phase Lock Gel Heavy Tubes. The upper RNA containing phase was precipitated by adding an equal volume of isopropanol and 0.1 vol of 5 M NaCl followed by incubation at room temperature for 10 min. The pellet was washed with 500 μl of 80% of ethanol and centrifuged (20,000 g, 5 min, 4°C). Biotinylated RNA was resuspended in 100 μl RNase-free water and final concentration of 1 μg/μl.

4sU-biotinylated RNA was separated from unlabeled RNA using streptavidin. RNA samples were denatured at 65°C for 10 min followed by rapid cooling on ice for 5 min. RNA was incubated with 200 μl of μMACs Streptavidin Microbeads (Miltenyi) for 15 min at room temperature while agitating. The columns were placed in an OctoMACS Seperator magnetic stand and were equilibrated with wash buffer (100 mM Tris-HCl pH 7.4, 10 mM EDTA, 1 M NaCl, 0.1% Tween 20). RNA samples with beads were loaded onto the column to enrich for 4sU-biotinylated RNA molecules. The flow through with mostly unlabeled RNA was reloaded twice onto the column. The column was washed 2 times with 200 μl of pre-warmed (55°C) wash buffer. The interactions between 4sU-biotinylated RNA and streptavidin beads were resolved by washing the column with 100 μl of 100 mM freshly prepared DTT in water. Elution steps were repeated with the addition of 100 μl of 100 mM DTT twice. Unlabeled and labeled RNA molecules were recovered by ethanol precipitation using 0.1 volume of 3 M NaOAc (pH 5.2) and 3 volumes of cold absolute ethanol. To recover labeled RNAs efficiently, 3 μl of glycogen (20 mg/ml, Thermo Fisher Scientific) was added prior to precipitation. The samples were incubated overnight at -20°C. Precipitated RNAs were collected by centrifugation at 20,000 g for 20 min at 4°C. The RNA pellet was rehydrated with an appropriate volume of RNase-free water and stored at -80°C. Total RNA concentration and quality of the samples were determined using an Agilent 2100 Bioanalyzer (Agilent Technologies) prior to generating the libraries for next generation sequencing.

## Library preparation and sequencing

Library preparation and sequencing was performed at the Heinrich Pette Institute, Hamburg, Germany. After isolation of total RNA its integrity was analyzed with the RNA 6000 Pico Chip on an Agilent 2100 Bioanalyzer (Agilent Technologies). Prior to library generation, RNA was

subjected to DNAse I digestion (Thermo Fisher Scientific) followed by RNeasy MinElute column clean up (Qiagen). RNA-Seq libraries were generated using the SMART-Seq v4 Ultra Low Input RNA Kit (Clontech Laboratories) according to the manufacturer´s recommendations. cDNA final libraries were generated utilizing the Nextera XT DNA Library Preparation Kit (Illumina). Concentrations of the final libraries were measured with a Qubit 2.0 Fluorometer (Thermo Fisher Scientific) and fragment lengths distribution was analyzed with the DNA High Sensitivity Chip on an Agilent 2100 Bioanalyzer (Agilent Technologies). All samples were normalized to 2 nM and pooled to become equimolar. The library pool was sequenced on the NextSeq500 (Illumina) with 2x150 bp (paired end), with 15.7 to 18.6 mio reads per sample.

## Bioinformatic analysis

The FastQ-sequences were analyzed with MultiQC (v1.7) [46] for quality issues and trimmed with Trimmomatic (0.39) [47] for leading and trailing bases with a quality threshold of 20 each. The reads were mapped to the Human Genome 19 with Hisat2 (2.1.0) [48] and file format conversion and sorting was done with Samtools (1.10) [49]. Since the targeted exon 2 of CD46 comprises 189 nucleotides, other exons with the same number of nucleotides were identified for comparison. The data from the genes comprising exons of 189 nts were extracted with Samtools' view command, indexed with Samtools' index, and the coverage for each base was calculated with Samtools' depth -a -r. Both normalization and visualization were done in R [50]. For normalization the samples were downsampled to the total number of mapped reads. The mean was applied to three independent replicates for visualization.

## Statistical analysis

All statistical analyses were performed using Prism v7.0 (GraphPad Software, San Diego, CA). Prior to test, graph kurtosis was analyzed to ensure normal or non-normal distribution. Based on the distribution analysis; Wilcoxon signed-rank test was used for comparisons between samples. Data are reported as mean and SEM.

## Supporting information

**S1 Fig. Molecular analysis of the *CD46* locus in CD46-Cas9 treated and non-treated B cells. (A)** Cellular DNAs were PCR amplified, sequenced and analyzed with the Outknocker webtool. The insertions and deletions in exon 2 of the *CD46* gene in WT EBV infected primary human B cells were analyzed 8 days post nucleofection. Reads from B cells nucleofected with the CD46-Cas9 RNP complexes (CD46-Cas9) were aligned to the hg19 reference human genome. The reads are summarized in pie charts and unique sequences are displayed below as examples. The target site of gRNA1, the upstream CD46 specific crRNA is highlighted in yellow within the reference sequence of the *CD46* locus; the PAM sequence is highlighted in orange. **(B)** Representative agarose gels of PCR products encompassing exon 2 of the *CD46* locus in non-infected and EBV-infected B cells after nucleofection with two CD46-Cas9 RNP complexes at indicated time point post nucleofection. The expected PCR product of the intact *CD46* locus is 276 bp in length. PCR products obtained from the *CD46* locus with the partially deleted exon 2 are about 178 bp in length. Experiments shown in panel B are representative of three independent biological replicates from non-infected and EBV infected cells.
(PDF)

**S2 Fig. Characterization of human B cell subsets purified from adenoid tissue and PBMCs and comparison of *CD46* knockout efficiencies. (A)** B cell identity after B cells isolation and

purification from adenoids or PBMCs **(B)** Flow cytometric analysis characterizing the different subsets of B cells purified from adenoid tissues or PBMC samples. The nomenclature describing different B cell subsets is depicted. SM, switched memory; N, naïve; UM, unswitched memory. **(C)** Flow cytometric analysis of CD46 surface levels on B cells obtained from adenoid and PBMC samples one week after nucleofection and infection with EBV. **(D)** B cells obtained from adenoid tissue were nucleofected with CD46-Cas9 RNP complexes and cultivated on irradiated CD40 ligand (CD40L) feeder cells in the presence of IL-4 for 9 days. As in panel C, the CD46 surface levels are shown.
(PDF)

**S3 Fig. Next generation sequencing of the *CDKN2A* locus in p16 KO and WT cells.** Cellular DNA of the *CDKN2A* locus was PCR amplified, sequenced and analyzed with the Outknocker webtool. The insertions and deletions in exon 1α of *CDKN2A* encoding p16$^{INK4a}$ were analyzed in primary human B cells two weeks post nucleofection. Reads for WT and p16 KO cells were aligned to the hg19 reference human sequence. The results are summarized in pie charts and unique read sequences are displayed below as examples. The target site of the *CDKN2A* specific crRNA and the PAM sequence are highlighted in yellow and orange, respectively, in the reference sequence.
(PDF)

**S4 Fig. p16$^{INK4a}$ is the only functional barrier to EBV driven proliferation of lymphoblastoid cells. (A)** To study the biological effect of the *CDKN2A* knockout in a time course analysis, p16 WT and p16 KO cells were mixed such that the fraction of the latter was in the order of 10 to 20% prior to infection as described in **Fig 4**. The graph shows the statistical analysis of the time-related frequencies of *CDKN2A* negative B cells (according to identified indels) infected with WT or ΔEBNA3C EBV strains from four independent experiments. Exemplary results from two experiments are shown in **Fig 4A**. Linear regression analysis was performed with data from all four experiments to calculate the slopes of linear regression from each experiment. **(B)** Slope values were transferred into a column data table. P values were calculated using the unpaired t-test. \*\*, P<0.05. **(C)** Cell expansion of four different B cell populations were recorded and plotted as a function of days post nucleofection (x-axis) versus the format of the cell culture vessel (y-axis) starting with a single well in a 48-well cluster plate up to T175 cell culture flasks. 2×10$^6$ B cells with an intact *CDKN2A* locus (p16 WT cells) or cells with an edited *CDKN2A* gene (p16 KO cells) were infected with the WT EBV (left panels) or ΔEBNA3C EBV (right panels). The results were consistent between four additional biological replicates. Three replicates are shown here and one is provided in **Fig 4C**.
(PDF)

**S5 Fig. CD138 expression in the RPMI-8226 cell line in comparison to an EBV-infected lymphoblastoid cell line (LCL) as a reference.** The left and middle panels show forward and sideward scatter data of an established EBV-infected lymphoblastoid cell line (LCL) and the RPMI-8226 cell line, respectively. Both cell lines were analyzed with a CD138 specific antibody (Invitrogen, APC-conjugated, #17-1389-42). RPMI-8226 cells (blue) express higher levels of CD138 than the LCL (red) in the graph on the right with normalized counts (y-axis). RPMI-8226 cells also reveal a subpopulation that exhibits a higher level of CD138 surface expression than the majority of the cells.
(PDF)

**S1 Table. Results of 182 differentially expressed genes after 4sU-RNA labeling, isolation of newly transcribed RNA and next generation sequencing of cDNA libraries.**
(DOCX)

**S2 Table. crRNA sequences.**
(PDF)

**S3 Table. List of primer pairs.**
(PDF)

## Acknowledgments

We would like to thank Christine Goebel for taking care of cells, occasionally. We also thank our colleagues Tanja Stief and Andreas Moosmann for teaching us how to propagate human primary B cells on CD40 ligand feeder cells. We thank Daniela Indenbirken and Adam Grundhoff, Heinrich Pette Institute Hamburg, for their support in establishing libraries from minute amounts of mRNA and sequencing them. We thank Andreas Brachmann, Biozentrum Ludwig-Maximilians-Universität München, and Veit Hornung, Gene Center, Ludwig-Maximilians-Universität München, for support in running the PCR library samples on their MiSeq instruments.

## Author Contributions

**Conceptualization:** Manuel Albanese, Wolfgang Hammerschmidt.

**Data curation:** Alexander Buschle.

**Formal analysis:** Ezgi Akidil, Manuel Albanese, Alexander Buschle.

**Funding acquisition:** Wolfgang Hammerschmidt.

**Investigation:** Ezgi Akidil, Manuel Albanese, Alexander Buschle, Adrian Ruhle, Dagmar Pich, Wolfgang Hammerschmidt.

**Methodology:** Ezgi Akidil, Manuel Albanese, Alexander Buschle, Adrian Ruhle.

**Project administration:** Wolfgang Hammerschmidt.

**Resources:** Oliver T. Keppler, Wolfgang Hammerschmidt.

**Software:** Ezgi Akidil, Manuel Albanese, Alexander Buschle.

**Supervision:** Alexander Buschle, Oliver T. Keppler, Wolfgang Hammerschmidt.

**Validation:** Ezgi Akidil, Manuel Albanese, Alexander Buschle, Wolfgang Hammerschmidt.

**Visualization:** Ezgi Akidil, Manuel Albanese, Alexander Buschle.

**Writing – original draft:** Ezgi Akidil, Manuel Albanese, Alexander Buschle, Oliver T. Keppler, Wolfgang Hammerschmidt.

**Writing – review & editing:** Ezgi Akidil, Manuel Albanese, Alexander Buschle, Wolfgang Hammerschmidt.

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
