## [Decision Letter · Decision Letter 0]

18 Dec 2020

Dear Prof. Hammerschmidt,

Thank you very much for submitting your manuscript "Highly efficient CRISPR-Cas9-mediated gene knockout in primary human B cells for functional genetic studies of Epstein-Barr virus infection" for consideration at PLOS Pathogens. As with all papers reviewed by the journal, your manuscript was reviewed by members of the editorial board and by several independent reviewers. The reviewers appreciated the attention to an important topic. Based on the reviews, we are likely to accept this manuscript for publication, providing that you modify the manuscript according to the review recommendations.

Overall, all three reviewers felt this manuscript to be a strong contribution and were interested in the methodology. Two of the reviewers had several specific points to address. In particular, reviewer #2 requests additional data on BLIMP1 levels in control vs CRISPR edited cells and clarification of the rational for using EBNA3C vs EBNA3A/3C doubly deficient EBV. Reviewer #1 has a number of constructive specific points to address.

Sincerely,

Benjamin E Gewurz, M.D., Ph.D.

Guest Editor

PLOS Pathogens

Erik Flemington

Section Editor

PLOS Pathogens

Kasturi Haldar

Editor-in-Chief

PLOS Pathogens

orcid.org/0000-0001-5065-158X

Michael Malim

Editor-in-Chief

PLOS Pathogens

orcid.org/0000-0002-7699-2064

Overall, all three reviewers felt this manuscript to be a strong contribution and were interested in the new methodological approach to CRISPR editing of freshly isolated primary human B-cells. While prior editing has been done in primary B-cells, published approaches have used stimulated B-cells that have been cultured for a period of time. Two of the reviewers had several specific points to address. In particular, reviewer #2 requests additional data on BLIMP1 levels in control vs CRISPR edited cells and clarification of the rational for using EBNA3C vs EBNA3A/3C doubly deficient EBV. Reviewer #1 has a number of constructive specific points to address.

Reviewer Comments (if any, and for reference):

Reviewer's Responses to Questions

**Part I - Summary**

Reviewer #1: In the work by Akidil and colleagues, a method was established to perform highly efficient CRISPR-Cas9-mediated gene knockout in primary human B cells. Nucleofection of B cells with preformed Cas9 ribonucleoprotein complexes was followed by EBV-mediated immortalization of the transfected B cells. Knockout of the CD46 gene was used as a model. This method was then applied to study an aspect of EBV infection of B cells. It is shown that EBNA3C has an important role in downregulating CDKN2A in lymphoblastoid cell lines, confirming earlier indications in this direction.

As primary B cells are notoriously difficult to genetically modify, the present novel protocol is a valuable tool to generate CRISPR-Cas9-modified cells. The manuscript is overall well written and the data are clearly presented.

The EBV infection of the cells as an intrinsic/mandatory part of the method is a restriction of the method and should be clearly mentioned already in the Abstract and Author summary.

Reviewer #2: The authors established a method and characterized its efficacy as well as kinetics to insert gene deletions via CRISPR/Cas9 in primary human B cells. They initially targeted CD46 with their knock-out technology and then CDKN2A, the gene for the CDK4 inhibitor p16INK4a. p16INK4a is in addition to BLIMP-1 one of the main targets for suppression by EBNA3C. Accordingly, the authors tested if p16INK4a deficient B cells got enriched during B cell transformation by deltaEBNA3C EBV infection towards lymphoblastoid cell lines (LCLs). This was indeed the case, but deltaEBNA3A/C deficient EBV did not induce significant plasma cell differentiation in the absence of p16INK4a. With these experiments the authors elegantly combined genetic host gene knock-outs and recombinant EBV mutants to conclusively address the dependence of EBV on EBNA3C mediated p16INK4a suppression during B cell transformation.

Even so p16INK4a had previously been identified as a target of EBNA3C, the reported experiments suggest p16INK4a inhibition as the main function of EBNA3C during B cell transformation by EBV, with CDKN2A knock-out cells getting enriched during deltaEBNA3C EBV infection to more than 80% in a competitive outgrowth assay. It would, however, be interesting to report what happens to the other main target BLIMP-1 in this EBNA3C deficient B cell transformation.

Reviewer #3: In this study, Akidil et al present an exciting technical development for efficient gene knockout in primary human B cells. They use CD46 as a test case initially and then move to more biologically relevant targets including the cyclin-dependent kinase inhibitor, p16INK4A. The innovation here is to use the Cas9/guide RNA ribonucleoprotein (RNP) complex to transfect primary resting human B cells followed by EBV infection to activate and induce proliferation of these cells. Initial studies support efficient knockout (KO) of the cell surface marker, CD46, which is easy to monitor by flow cytometry. They find that one guide is capable of ~60% KO by flow cytometry and sequencing of the target genomic DNA site while two guides approaches 80% KO. Temporal analysis of allele targeting indicates progressive error-prone repair through the first 24h after transfection with the highest indel frequencies reported at 4 days post transfection and persistent through 8 days. Following the fate of the RNAs in these (CD46 sg1+2) cells indicate about 50% of the CD46 transcripts contain indels or deletions. However, this transfection did not significantly impact B cell gene expression, as anticipated.

To follow up with a biologically relevant target, the authors transfect RNPs against p16INK4a, which the viral EBNA3 proteins are known to antagonize. Excitingly, this novel approach recapitulated the findings by the Allday group indicating that p16 loss enabled outgrowth of B cells infected with an EBNA3C deleted EBV. The follow up to this experiment was a test of the recent hypothesis that the loss of both EBNA3A and 3C leads to a plasma cell phenotype with increased expression of CD138 (proxy for antibody secretion) and CD38 (activation marker). The data presented support this hypothesis, though the phenotype was not as strong as shown by Styles, et al. There are technical challenges associated with observing a true differentiation to the plasma cell lineage and antibody secretion. For example, these cells will likely cease to proliferate. However, with p16 knocked-out, the authors conclude that this terminal differentiation of 3A/3C KO infected B cells does not occur. It may be of interest to determine Ig expression and secretion is increased in these p16KO, EBNA3A/3C KO cells at 20 days (or earlier) post infection.

**Part II – Major Issues: Key Experiments Required for Acceptance**

Reviewer #1: No additional experiments requested.

Reviewer #2: 1. The authors report minimal plasma cell surface marker up-regulation during CDKN2A deficient B cell transformation. It would be interesting to report the BLIMP-1 levels in the knock-out and mutant infected B cells. BLIMP-1 is considered as one of the main transcription factors for plasma cell differentiation.

2. The authors seem to have used EBNA3A and EBNA3C double-deficient EBV for the transformation of CDKN2A deficient B cells to investigate plasma cell differentiation, while they used EBNA3C deficient EBV for the competitive wt versus CDKN2A knock-out infection experiments. Is there a particular reason for this? Would plasma cell differentiation be enhanced in the presence of EBNA3A?

3. It would be good if a summary graph could be provided for the four experiments of figures 4B and S3A with statistical evaluation. Maybe enrichment of CDKN2A indel B cells in deltaEBNA3C versus wt EBV infection could be compared at select timepoints.

Reviewer #3: (No Response)

**Part III – Minor Issues: Editorial and Data Presentation Modifications**

Reviewer #1: a) A major aspect of the method is that one day after nucleofection the cells are infected with EBV to immortalize them. Thus, although the experiment is indeed started with primary B cells, one later obtains EBV-immportalized lymphoblastoid cell lines. As EBV has a massive influence on B cell physiology and gene expression, this is an important aspect/caveat of the method. Therefore, the EBV infection of the cells as an intrinsic/mandatory part of the method should be clearly mentioned already in the Abstract and Author summary.

b) From the Methods section it becomes obvious that the studies were not performed with pure B cells, but with tonsillar mononuclear cells that were depleted from T cells by erythrocyte rosetting and cyclosporin A treatment. Hence, the resulting cells likely also encompass monocytes, NK cells, and other cells present in tonsils. Did the authors test the B cell fraction in the suspensions they used? Could it be that the non-B cells present in the suspensions used could have a supporting role in the analysis? This would be relevant to know if others plan to use the method for isolated, pure B cells.

c) The cells from adenoids that were used encompass naive B cells, memory B cells, germinal center B cells, and a few transitional and CD5-positive mature B cells. Do the authors have any data whether all of these B cell subsets are well nucleofected and genetically modified? As germinal center B cells activate an apoptosis program once they are in suspension, it would be interesting to know whether also these cells can be modified and rescued by the approach.

d) Although the method as such is novel, the authors should also refer to and briefly discuss two other recent methods for CRISPR-Cas9-mediated gene knockout in primary human B cells: R. Caeser et al., Nature Commun 2019 (focussed on germinal center B cells), M.J. Johnson et al., Scientific Reports, 2018.

e) The authors used cyclosporin A in the initial culture. This was likely done to suppress T cells. However, as cyclosporin A affects NFAT signaling, which has also been reported to occur in some circumstances in B cells, did the authors consider potential effects of cyclosporin A on subsets among their mixed B cell populations?

f) The numbering of the supplementary tables is not in chronological order.

g) The oligonucleotide sequences in supplementary tables 1 and 2 should be provided with information on their orientation (mark 5‘ and 3‘ ends).

h) In the legend to Supplementary figure S3, the reference to main figure 3 is likely wrong.

Reviewer #2: 1. The timepoint of analysis after EBV infection could be added for clarity to figure 1A.

Reviewer #3: (No Response)

PLOS authors have the option to publish the peer review history of their article (what does this mean?). If published, this will include your full peer review and any attached files.

Reviewer #1: **Yes: **Ralf Küppers

Reviewer #2: No

Reviewer #3: **Yes: **Micah Luftig
---

## [Editor Report · Decision Letter 1]

4 Mar 2021

Dear Prof. Hammerschmidt,

We are pleased to inform you that your manuscript 'Highly efficient CRISPR-Cas9-mediated gene knockout in primary human B cells for functional genetic studies of Epstein-Barr virus infection' has been provisionally accepted for publication in PLOS Pathogens.

Best regards,

Benjamin E Gewurz, M.D., Ph.D.

Guest Editor

PLOS Pathogens

Erik Flemington

Section Editor

PLOS Pathogens

Kasturi Haldar

Editor-in-Chief

PLOS Pathogens

orcid.org/0000-0001-5065-158X

Michael Malim

Editor-in-Chief

PLOS Pathogens

orcid.org/0000-0002-7699-2064
---

## [Editor Report · Acceptance letter]

24 Mar 2021

Dear Prof. Hammerschmidt,

We are delighted to inform you that your manuscript, "Highly efficient CRISPR-Cas9-mediated gene knockout in primary human B cells for functional genetic studies of Epstein-Barr virus infection," has been formally accepted for publication in PLOS Pathogens.

Best regards,

Kasturi Haldar

Editor-in-Chief

PLOS Pathogens

orcid.org/0000-0001-5065-158X

Michael Malim

Editor-in-Chief

PLOS Pathogens

orcid.org/0000-0002-7699-2064